# HALLUCINATING LLM COULD BE CREATIVE

## ABSTRACT

Large Language Models (LLMs) frequently produce hallucinations, which typically refer to the information that appears reasonable but is false or inaccurate generated by LLMs. On the other hand, hallucinations aren't entirely negative. Exploring the notion of **good hallucinations** that may contribute to **creativity and innovation** in LLMs. We propose a new metric to assess the quality of creativity in hallucination, focusing on correctness, consistency, and reasoning diversity. We sampled LLM's responses many times and used semantic clustering to indicate the good hallucination sample, trying to evaluate the responses using our proposed metric. Our experiments explore different prompting strategies and hyperparameter configurations, providing comprehensive results based on these metrics to investigate their impact on creativity. Preliminary results show that LLMs can generate creative responses from hallucinations while maintaining a low rate of factual errors. This research offers a more fine-grained and unique perspective on hallucinations in LLMs, proposing a possible strategy to harness the creative potential in hallucinations to raise awareness that hallucinations are not necessarily an absolutely negative phenomenon.

*"Maybe hallucinations are just another reality that we don't see most of the time."*

*− Lynne Ewing*

## 1 INTRODUCTION

Large Language Models (LLMs), such as GPT-4o and Claude 3.5 Sonnet, have significantly advanced the field of Natural Language Processing (NLP) by demonstrating remarkable capabilities in generating human-like text (Chen et al., 2024; 2023), performing complex language tasks (Wei et al., 2022), and engaging in coherent dialogues. These models have been instrumental across various domains, including content creation, code synthesis, education, and conversational agents, transforming academic research and industrial applications (Shen et al., 2023; Ge et al., 2023).

Despite these advancements, the persistent challenge of hallucinations in LLMs remains a significant concern. Hallucinations refer to factually incorrect, logically inconsistent, or nonsensical outputs, yet sometimes appear to be true (Farquhar et al., 2024). Such outputs are typically unwelcome, especially in critical applications like legal consultation, medical advice, or scientific research, where accuracy and reliability are of utmost importance (Gunjal et al., 2024). As a result, significant efforts have been made to mitigate hallucinations through fine-tuning with accurate datasets, improving output consistency, and using post-processing methods like fact-checking (Guan et al., 2024).

We propose an emerging perspective that suggests not all hallucinations are entirely detrimental. ***In contexts that emphasize creativity and innovation, hallucinations can contribute positively by introducing novel ideas or unconventional reasoning paths.*** Specifically, when a LLM arrives at a correct answer through an alternative reasoning process, it may offer valuable insights to foster creative problem-solving ability in a special way. After presenting this perspective, we further raise the research question in our study: *Can certain hallucinations, instead of being purely negative, play a constructive role in fostering creativity while ensuring accuracy and maintaining correctness?*

In this study, we explore the notion of **good hallucinations**—instances where LLMs generate different reasoning paths that still lead to correct answers. Examining these alternative and dynamic reasoning processes, we aim to understand how LLMs can contribute to creative thinking without compromising factual accuracy. The Torrance Tests of Creative Thinking (TTCT) is the most widely

used and extensively researched creativity assessment across fields like psychology, with substantial data supporting its reliability and validity (Torrance, 1966; Hass et al., 2016). To this end, we propose metrics to evaluate the quality of these good hallucinations, focusing on accuracy, coherence, and reasoning diversity, which closely align with the TTCT in psychology (Zhao et al., 2024).

We evaluated multiple times generated responses and employed semantic clustering techniques (Randriamihamison et al., 2021) to quantify the diversity and creativity of the reasoning paths. Our research explores various prompting techniques and hyperparameter configurations based on psychological creativity metrics to understand their impact on generating creative yet accurate outputs. Specifically, we examine the impact of different prompting strategies and settings, such as temperature adjustments to change certainty, on the diversity and correctness of the generated reasoning paths to determine which approach better balances creative freedom with factual precision.

- **Introduction of the concept *good hallucination*:** We define and explore the *good hallucinations* that, despite diverging in reasoning paths, contribute to creativity without sacrificing accuracy.
- **Development of metrics for quality assessment of good hallucination:** We propose metrics focusing on correctness, consistency, and reasoning diversity, align with the TTCT in psychology. Semantic clustering techniques are utilized to evaluate these metrics quantitatively.
- **Experimental analysis of prompting techniques and hyperparameters:** We conduct extensive experiments to understand how different prompting strategies and hyperparameter settings like temperature adjusting influence the generation of creative yet accurate outputs in LLMs.

This research highlights these positive aspects of hallucinations and offers a refined perspective on their role in LLMs' outputs. We hope strategies to leverage **good hallucinations** in applications that benefit from creative problem-solving, thereby enhancing the flexibility and utility of AI systems.

## 2 METHODOLOGY

Our methodology evaluates the creativity of Large Language Models (LLMs) by introducing a novel metric that combines both the accuracy and the diversity of their generated reasoning paths. This is achieved by sampling multiple outputs from the LLM, clustering them based on semantic similarity, and calculating a composite creativity score that encapsulates both correctness and variety in reasoning, as illustrated in Figure 1.

### 2.1 SEMANTIC CLUSTERING FOR REASONING PATHS

To assess the diversity of reasoning paths, we generate multiple outputs from the LLM and cluster these outputs using two different methods: (1) clustering based on text embeddings, and (2) clustering via LLM-based prompting.

#### 2.1.1 GENERATING MULTIPLE OUTPUTS

We prompt the LLM with a specific question or task and generate $N$ different reasoning paths by sampling its output multiple times using different decoding methods. Formally, let:

$$\mathcal{R}_{\text{all}} = \{r_1, r_2, \ldots, r_N\},$$

represent the complete set of reasoning paths generated by the LLM, where each $r_i$ corresponds to an individual reasoning path responding to the prompt $P$.

Given that incorrect responses introduce potential noise and do not contribute meaningfully to the assessment of creativity, we exclude all reasoning paths leading to incorrect final answers. This filtering process ensures that only valid reasoning paths are considered for subsequent analysis.

We define the filtered set of reasoning paths $\mathcal{R}$ as:

$$\mathcal{R} = \{r_i \in \mathcal{R}_{\text{all}} \mid \text{Answer}(r_i) = \text{Correct Answer}\},$$

The number of correct reasoning paths is then $N_{\text{correct}} = |\mathcal{R}|$.

By restricting our focus to correct reasoning paths, we effectively mitigate the impact of incorrect or potentially misleading outputs, thereby enhancing the robustness of our creativity analysis. This

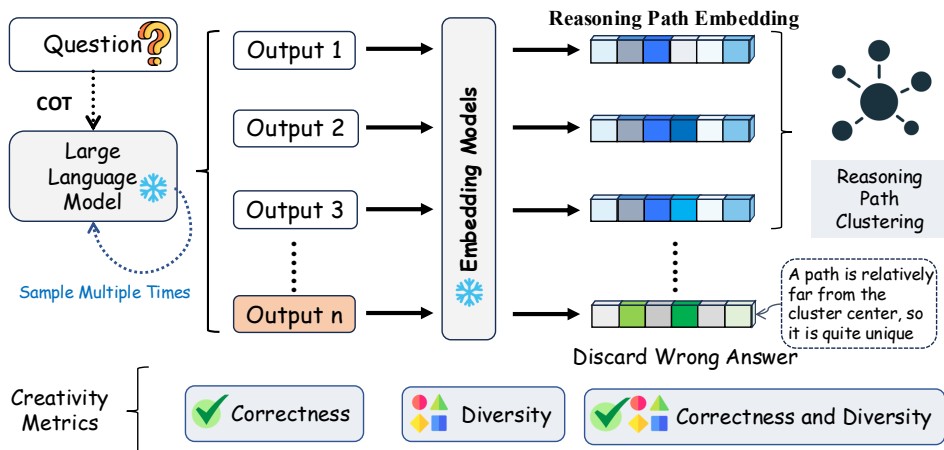

Figure 1. An overview of the proposed metric for evaluating LLM creativity. The process involves generating multiple reasoning paths, transforming them into embeddings, clustering based on semantic similarity, and computing a composite creativity score that combines accuracy and diversity metrics.

approach is consistent with the underlying premise that creative reasoning holds value primarily when it culminates in accurate and valid conclusions.

### 2.1.2 METHOD 1: CLUSTERING VIA TEXT EMBEDDINGS

In this approach, each reasoning path $r_i \in \mathcal{R}$ is mapped to a high-dimensional vector representation through a pre-trained text embedding model $\phi$. The embeddings for each reasoning path are expressed as:

$$e_i = \phi(r_i), \quad \forall i \in \{1, 2, \ldots, N_{\text{correct}}\},$$

We perform clustering on the set of embeddings $\mathcal{E} = \{e_1, e_2, \ldots, e_{N_{\text{correct}}}\}$ using agglomerative hierarchical clustering with a specified distance threshold $d_{\text{thresh}}$. The process begins by treating each embedding as an individual cluster, then iteratively merges the two closest clusters based on a chosen distance metric, such as Euclidean distance or cosine similarity. Clustering continues until the distance between all remaining clusters exceeds the threshold $d_{\text{thresh}}$.

This method allows clusters to form based on intrinsic similarities among the reasoning paths without predefining the number of clusters. The set of clusters obtained is denoted as $\mathcal{C} = C_1, C_2, \ldots, C_K$, where $K$ is determined by the clustering algorithm based on the data and the specified threshold.

### 2.1.3 METHOD 2: CLUSTERING VIA LLM PROMPTING

Alternatively, we leverage the LLM's understanding to cluster the reasoning paths directly through a specially designed prompt. We construct a prompt that instructs the LLM to classify the different reasoning paths and return the clusters in a structured format.

**An example prompt is:**
```
You are a helpful assistant that can classify different
reasoning paths.  Given the following reasoning paths, please
group them based on their underlying reasoning strategies and
return the clusters in JSON format.
```

We input the filtered set of reasoning paths $\mathcal{R}$ into the LLM along with the above prompt. The LLM processes this information and outputs the clusters $\mathcal{C} = C_1, C_2, \ldots, C_K$, where each cluster contains reasoning paths that share similar strategies.

### 2.1.4 ALGORITHM

The complete process for both methods, including the filtering step and hierarchical clustering, is outlined in Algorithm 1.

---

**Algorithm 1** Semantic Clustering of LLM Responses

---

**Require:** LLM model $\mathcal{M}$, prompt $P$, number of samples $N$, embedding function $\phi$, distance threshold $d_{\text{thresh}}$ (for Method 1), clustering prompt $P_{\text{cluster}}$ (for Method 2)
**Ensure:** Clusters of reasoning paths $\mathcal{C} = C_1, C_2, \ldots, C_K$
 1: **Generate reasoning paths:**
 2: Initialize $\mathcal{R}\text{all} \leftarrow \emptyset$
 3: **for** $i = 1$ to $N$ **do**
 4:     Generate reasoning path $r_i \leftarrow \mathcal{M}(P)$ with stochastic sampling
 5:     $\mathcal{R}\text{all} \leftarrow \mathcal{R}\text{all} \cup r_i$
 6: **end for**
 7: **Filter out incorrect answers:**
 8: $\mathcal{R} \leftarrow r_i \in \mathcal{R}\text{all} \mid \text{Answer}(r_i) = \text{Correct Answer}$
 9: **if** Using Method 1 (Hierarchical Clustering with Distance Threshold) **then**
10:     Compute embeddings $\mathcal{E} \leftarrow \phi(r_i) \mid r_i \in \mathcal{R}$
11:     Perform agglomerative hierarchical clustering on $\mathcal{E}$ with distance threshold $d_{\text{thresh}}$ to obtain clusters $\mathcal{C}$
12: **else if** Using Method 2 (LLM Prompting) **then**
13:     Construct clustering prompt $P_{\text{cluster}}$ including reasoning paths $\mathcal{R}$
14:     Obtain clusters $\mathcal{C} \leftarrow \mathcal{M}(P_{\text{cluster}})$
15: **end if**

---

## 2.2 CREATIVITY METRIC

We propose a novel creativity metric that combines accuracy and diversity into a unified score. This metric is meticulously designed to reflect both the correctness of the responses and the variety in the reasoning paths generated by the LLM, thereby providing a comprehensive assessment of the model's creative capabilities.

### 2.2.1 ACCURACY COMPONENT

The accuracy component $A$ is defined as the proportion of generated responses that are correct:

$$A = \frac{1}{N} \sum_{i=1}^{N} \mathbb{I}\big[\text{Answer}(r_i) = \text{Correct Answer}\big]$$

where $N$ is the total number of generated responses, $\mathbb{I}[\cdot]$ is the indicator function, and $\text{Answer}(r_i)$ extracts the final answer from reasoning path $r_i$. This component effectively measures the LLM's ability to produce correct answers, which is a fundamental aspect of its performance.

### 2.2.2 DIVERSITY COMPONENT

The diversity component, denoted as $D$, measures the spread and balance of reasoning paths across different clusters obtained from the semantic clustering process. It is calculated using the normalized entropy of the cluster distribution:

$$D = -\frac{1}{\log K} \sum_{k=1}^{K} p_k \log p_k$$

where $K$ is the total number of clusters resulting from the clustering algorithm applied to the correct responses, $p_k$ represents the proportion of correct responses in cluster $C_k$, defined as:

$$p_k = \frac{|C_k|}{N_{correct}}$$

where $|C_k|$ is the number of correct responses in cluster $C_k$ and $N_{\text{correct}}$ is the number of correct responses after filtering out incorrect answers. This formulation ensures that $D \in [0, 1]$, with higher values indicating greater diversity among the reasoning paths.

### 2.2.3 Unified Creativity Score

The unified creativity score $C$ synthesizes the accuracy and diversity components into a single scalar value using a weighted geometric mean:

$$C = (A + \epsilon)^\alpha \times (D + \epsilon)^{1-\alpha}$$

where $\alpha \in [0, 1]$ controls the trade-off between accuracy and diversity, and $\epsilon$ is a small constant added to prevent multiplication by zero.

### 2.2.4 Algorithm

The procedure for computing the creativity score is detailed in Algorithm 2.

---

**Algorithm 2** Computation of the Creativity Score

---

**Require:** Set of reasoning paths $\mathcal{R}$all, total number of responses $N$, weighting parameter $\alpha$, small constant $\epsilon$
**Ensure:** Creativity score $C$
1: **Filter correct responses:**
2: $\mathcal{R} \leftarrow r_i \in \mathcal{R}$all $\mid$ Answer$(r_i) = $ Correct Answer
3: $N_{\text{correct}} \leftarrow |\mathcal{R}|$
4: **Compute accuracy:**
   $A \leftarrow \frac{N_{\text{correct}}}{N}$
5: **Cluster correct reasoning paths to obtain** $\mathcal{C} = C_1, C_2, \ldots, C_K$
6: **Compute cluster proportions:**

$$p_k \leftarrow \frac{|C_k|}{N_{\text{correct}}}, \quad \forall k \in \{1, 2, \ldots, K\}$$

7: **Compute diversity:**
   $D \leftarrow -\frac{1}{\log K} \sum_{k=1}^{K} p_k \log p_k$
8: **Compute creativity score:**
   $C \leftarrow (A + \epsilon)^\alpha \times (D + \epsilon)^{1-\alpha}$

---

### 2.2.5 Interpretation and Parameter Selection

The creativity score $C$ allows for balancing accuracy and diversity through the parameter $\alpha$. When $\alpha = 1$, $C$ focuses purely on accuracy. At $\alpha = 0$, the score shifts to prioritize diversity among correct responses. For intermediate values of $\alpha$, the score incorporates both accuracy and diversity, with the specific weight determined by $\alpha$. For example, choosing $\alpha = 0.75$ would prioritize accuracy while still considering diverse reasoning strategies.

### 2.3 Implementation Details

For each prompt $P$, we generated 15 reasoning paths to maintain computational feasibility while capturing diverse strategies. Using the baseline LLM, we applied nucleus sampling with a top-$p$ (Holtzman et al., 2020) value of 0.9 to introduce variability without sacrificing coherence. The clustering approach used agglomerative hierarchical clustering (Randriamihamison et al., 2021) with cosine distance. We set a threshold of 0.3 for merging clusters, determined through preliminary experiments to balance clustering granularity.

## 3 Experiments

We employed a diverse range of datasets to comprehensively assess the performance of various Large Language Models (LLMs), such as Llama-3 (Dubey et al., 2024), across different reasoning

| LLM | GSM8k | | | | | MultiArith | | | | |
|---|---|---|---|---|---|---|---|---|---|---|
| | Acc | Div | $\alpha = 0.25$ | $\alpha = 0.5$ | $\alpha = 0.75$ | Acc | Div | $\alpha = 0.25$ | $\alpha = 0.5$ | $\alpha = 0.75$ |
| Llama 3-8B | 77.26 | 73.56 | 72.77 | 72.53 | 72.79 | 96.18 | 85.53 | **87.48** | **89.65** | **92.08** |
| Llama 3.1-8B | 82.93 | **81.26** | 80.66 | 80.50 | 80.73 | 59.71 | **86.32** | 76.81 | 69.53 | 63.85 |
| Mistral 0.2-7B | 44.90 | 58.91 | 52.66 | 48.08 | 44.73 | 71.92 | 79.18 | 75.41 | 72.79 | 71.06 |
| Qwen 2.5-7B | **88.27** | 81.10 | **81.79** | **82.81** | **84.14** | **98.84** | 83.80 | 86.34 | 89.06 | 92.03 |

Table 1. Main results on math reasoning datasets. The largest value in each column is in bold, and the second largest is underlined.

tasks. These datasets are divided into three categories: mathematical reasoning, creative problem-solving, and commonsense and strategic question-answer tasks. Below, we introduce each dataset and provide a brief summary in Section 3.1 of its task and purpose within its respective category.

## 3.1 DATASET DESCRIPTION

### 3.1.1 MATHEMATICAL REASONING DATASETS

**GSM8K:** The Grade School Math 8K (GSM8K) dataset (Cobbe et al., 2021) comprises 8,500 high-quality grade school math word problems. Each problem requires the solver to perform multi-step reasoning and numerical calculations to arrive at the correct solution.

**MultiArith:** MultiArith (Kojima et al., 2023) is a dataset containing multi-step arithmetic word problems. These problems involve sequential numerical reasoning, where multiple arithmetic operations must be executed in the correct order to solve the problem.

These datasets assess the LLM's ability to understand mathematical concepts and perform precise arithmetic operations.

### 3.1.2 CREATIVE PROBLEM-SOLVING DATASETS

**RiddleSense**: RiddleSense (Lin et al., 2021) is a dataset featuring challenging riddles that demand creative thinking and inferential reasoning. The riddles often include figurative language, analogies, and double meanings, requiring abstract interpretation beyond literal comprehension.

**MacGyver**: The MacGyver (Tian et al., 2024) dataset explores the creative problem-solving capabilities of modern LLMs in a novel constrained setting. It consists of over 1,600 real-world problems deliberately designed to trigger innovative usage of objects and necessitate out-of-the-box thinking. The tasks require the model to devise creative solutions using limited resources, focusing on intricate aspects of physical reasoning, planning, and unconventional thinking.

These datasets evaluate the LLM's capacity for creative thinking and its ability to generate innovative solutions to unconventional problems.

### 3.1.3 COMMONSENSE AND STRATEGIC REASONING DATASETS

**StrategyQA**: StrategyQA (Geva et al., 2021) is a question-answering dataset where each question necessitates implicit reasoning steps and strategic thinking. The model must construct multi-hop reasoning paths and apply strategic knowledge to derive the correct answers.

**CommonsenseQA**: CommonsenseQA (Talmor et al., 2019) is a multiple-choice question-answering dataset designed to test the model's ability to utilize commonsense knowledge. The questions cover a wide range of everyday situations, requiring the model to select the most plausible answer based on general world understanding.

These datasets test the LLM's understanding of commonsense knowledge and its ability to perform strategic, multi-hop reasoning.

| LLM | RiddleSense | | | | | MacGyver | | | | |
|-----|------|------|------------------|-----------------|------------------|------|------|------------------|-----------------|------------------|
| | Acc | Div | $\alpha = 0.25$ | $\alpha = 0.5$ | $\alpha = 0.75$ | Acc | Div | $\alpha = 0.25$ | $\alpha = 0.5$ | $\alpha = 0.75$ |
| Llama 3-8B | 43.21 | 81.76 | 67.68 | 57.09 | 48.99 | 43.67 | 40.44 | 41.12 | 41.82 | 42.55 |
| Llama 3.1-8B | 59.89 | **86.20** | **76.86** | **69.68** | 64.05 | **46.34** | **43.13** | **45.42** | **45.73** | **46.03** |
| Mistral 0.2-7B | 50.77 | 71.10 | 62.99 | 57.26 | 53.16 | 37.61 | 34.72 | 35.31 | 35.93 | 36.58 |
| Qwen 2.5-7B | **64.84** | 76.65 | 71.94 | 68.53 | **66.05** | 45.26 | 42.21 | 42.95 | 43.71 | 44.48 |

Table 2. Main results on creative problem-solving datasets.

| LLM | CommonsenseQA | | | | | StrategyQA | | | | |
|-----|------|------|------------------|-----------------|------------------|------|------|------------------|-----------------|------------------|
| | Acc | Div | $\alpha = 0.25$ | $\alpha = 0.5$ | $\alpha = 0.75$ | Acc | Div | $\alpha = 0.25$ | $\alpha = 0.5$ | $\alpha = 0.75$ |
| Llama 3-8B | 41.15 | 78.75 | 64.87 | 54.51 | 46.65 | 62.09 | 77.51 | 71.51 | 67.05 | 63.72 |
| Llama 3.1-8B | 69.63 | **85.38** | 79.76 | 75.40 | 71.97 | **73.53** | **82.54** | **78.54** | **75.71** | **73.77** |
| Mistral 0.2-7B | 64.19 | 77.97 | 72.50 | 68.56 | 65.70 | 57.65 | 69.02 | 63.93 | 60.40 | 58.00 |
| Qwen 2.5-7B | **80.87** | 81.58 | **80.41** | **79.83** | **79.71** | 70.50 | 73.79 | 71.51 | 70.07 | 69.27 |

Table 3. Main results on commonsense and strategic datasets. The largest value in each column is in bold, and the second largest is underlined.

## 3.2 MAIN EXPERIMENT

In this section, we present the core experiments conducted to evaluate the performance of the selected Large Language Models (LLMs) across various tasks. The primary objective is to assess the models' overall capabilities using the established creativity metric.

For each dataset, we generated multiple outputs per prompt. Specifically, we sampled 15 reasoning paths for each question to capture the diversity of possible solutions. The models were evaluated using the proposed creativity metric, which combines accuracy and diversity into a unified score.

**Results.** As shown in Tables 1, 2, and 3, Qwen 2.5-7B (Hui et al., 2024) achieves the highest accuracy in math reasoning (GSM8K: 88.27%, MultiArith: 98.84%) and commonsense tasks (CommonsenseQA: 80.87%), highlighting its strength in precise, knowledge-driven tasks. Llama 3.1-8B (Dubey et al., 2024) excels in strategic reasoning and creative problem-solving, scoring highest in StrategyQA (73.53%) and MacGyver (46.34%), while also leading in diversity on RiddleSense (86.20%). Llama 3-8B and Mistral 0.2-7B (Jiang et al., 2023) perform competitively but generally lower, with Mistral 0.2-7B trailing in accuracy across most datasets. The results show that Qwen 2.5-7B and Llama 3.1-8B stand out for their strong performance across different reasoning and problem-solving tasks.

## 3.3 HYPERPARAMETER SENSITIVITY

In this subsection, we explore how different hyperparameter settings impact the outputs of the Large Language Models (LLMs). Specifically, we focus on two key parameters: the temperature in the sampling process and the inference strategies employed during text generation. We conduct experiments using the Llama 3.1-8B model on the GSM8K and StrategyQA datasets to evaluate the effects of these parameters on creativity and accuracy.

### 3.3.1 EFFECT OF TEMPERATURE ON MODEL PERFORMANCE

The temperature parameter in the sampling process controls the randomness of the LLM's output. Lower temperatures make the model's output more deterministic, favoring high-probability tokens, while higher temperatures increase randomness, allowing for more diverse but potentially less coherent outputs. We varied the temperature parameter over the values 0.1, 0.5, 0.7, and 0.9, generating multiple reasoning paths for the prompts in the GSM8K and StrategyQA datasets at each setting. We then computed the accuracy (Acc), diversity (Div), and unified creativity scores at different values of the weighting parameter $\alpha$ to evaluate the performance across these settings.

**Results.** As shown in Table 4, the temperature setting significantly affects both accuracy and diversity across the GSM8K and StrategyQA datasets. A temperature of 0.6 achieves the highest diversity on GSM8K (82.16%) while maintaining a strong accuracy of 84.57%. Conversely, a temperature of 0.8 yields the highest accuracy on StrategyQA (73.53%) and the highest diversity (82.54%). Lower

| Llama 3.1-8B | GSM8K | | | | | StrategyQA | | | | |
|---|---|---|---|---|---|---|---|---|---|---|
| | Acc | Div | $\alpha = 0.25$ | $\alpha = 0.5$ | $\alpha = 0.75$ | Acc | Div | $\alpha = 0.25$ | $\alpha = 0.5$ | $\alpha = 0.75$ |
| $t = 0.2$ | 84.17 | 66.99 | 67.26 | 67.84 | 68.86 | 69.90 | 72.34 | 70.27 | 68.97 | 68.27 |
| $t = 0.4$ | **84.80** | 76.80 | 76.87 | 77.31 | 78.13 | 69.00 | 73.93 | 70.55 | 68.30 | 66.90 |
| $t = 0.6$ | 84.57 | **82.16** | **81.74** | **81.85** | **82.39** | 70.63 | 75.09 | 72.09 | 70.16 | 69.03 |
| $t = 0.8$ | 82.93 | 81.26 | 80.66 | 80.50 | 80.73 | **73.53** | **82.54** | **78.54** | **75.71** | **73.77** |

Table 4. Effects of different temperature settings on the Llama 3.1-8B model's performance on the GSM8K and StrategyQA datasets. The largest value in each column is in bold, and the second largest is underlined.

| Llama 3.1-8B | GSM8K | | | | | StrategyQA | | | | |
|---|---|---|---|---|---|---|---|---|---|---|
| | Acc | Div | $\alpha = 0.25$ | $\alpha = 0.5$ | $\alpha = 0.75$ | Acc | Div | $\alpha = 0.25$ | $\alpha = 0.5$ | $\alpha = 0.75$ |
| Greedy | 84.17 | 66.99 | 67.26 | 67.84 | 68.86 | 69.90 | 72.34 | 70.27 | 68.97 | 68.27 |
| Top-$k = 50$ | 82.93 | **81.26** | **80.66** | **80.50** | **80.73** | 73.53 | **82.54** | **78.54** | **75.71** | **73.77** |
| Top-$p = 0.95$ | 84.73 | 79.54 | 79.08 | 79.27 | 80.03 | 69.90 | 72.34 | 70.27 | 68.97 | 68.27 |
| DoLa | **86.45** | 76.12 | 78.58 | 81.12 | 83.74 | **75.89** | 78.24 | 77.64 | 77.05 | 76.47 |

Table 5. Effects of different decoding methods on the Llama 3.1-8B model's performance on the GSM8K and StrategyQA datasets. The largest value in each column is in bold, and the second largest is underlined.

temperatures (0.2 and 0.4) result in higher accuracy but lower diversity, indicating more deterministic and less varied outputs. These results demonstrate a trade-off between accuracy and diversity controlled by the temperature parameter, suggesting that selecting an optimal temperature depends on the specific requirements of the task, whether prioritizing precision or creativity.

### 3.3.2 Comparison of Inference Strategies

The choice of inference strategy during text generation plays a crucial role in shaping the quality, diversity, and creativity of the model's outputs, extending beyond the impact of tuning parameters such as temperature. This section compares standard decoding methods with two advanced strategies recently introduced in the literature.

**Standard Decoding Methods** Greedy search are commonly employed decoding techniques. Greedy search deterministically selects the token with the highest probability at each step. While this approach is efficient, it often leads to homogeneous outputs that lack diversity.

To introduce variability and enhance the diversity of generated content, stochastic methods like **Top-$k$ sampling** (Fan et al., 2018) and **Top-$p$ (nucleus) sampling** (Holtzman et al., 2020) are used. Top-$k$ sampling limits the candidate tokens to the $k$ most probable options at each decoding step and samples from this subset. This method prevents the model from considering low-probability tokens, reducing irrelevant or nonsensical outputs while still allowing for diversity.

Top-$p$ sampling, on the other hand, selects tokens from the smallest possible set whose cumulative probability exceeds a predefined threshold $p$. This dynamic approach adjusts the candidate pool based on the distribution of the probabilities, providing a balance between diversity and coherence. Both Top-$k$ and Top-$p$ sampling aim to mitigate the shortcomings of greedy search by avoiding deterministic and repetitive outputs, thereby improving the overall quality of the generated text.

**Advanced Decoding Strategies** Recent advancements in decoding methods have introduced more sophisticated strategies to enhance the quality of generated content. **Decoding by Contrasting Layers** (DoLa), introduced by Chuang et al. (2024), leverages the internal structure of transformer models by contrasting logits from deeper and earlier layers. In our experiments, we use layer 32 as the mature layer and layers $0, 2, 4, 6, 8, 10, 12, 14$ as candidate premature layers. This technique, based on the observation that factual knowledge tends to concentrate in specific layers, emphasizes such layers to reduce hallucinations and improve the factual accuracy of generated content. These strategies were tested using the Llama 3.1-8B model, generating multiple reasoning paths per prompt and evaluated for accuracy, diversity, and creativity across the GSM8K and StrategyQA datasets.

**Results.** Table 5 shows that decoding strategy significantly impacts accuracy and diversity on GSM8K and StrategyQA. **DoLa** achieves the highest accuracy, with 86.45% on GSM8K and 75.89% on StrategyQA, surpassing greedy search and stochastic methods. **Top-$k$ sampling** ($k = 50$) yields

| Llama 3.1-8B | GSM8k | | | | | StrategyQA | | | | |
|---|---|---|---|---|---|---|---|---|---|---|
| | Acc | Div | $\alpha = 0.25$ | $\alpha = 0.5$ | $\alpha = 0.75$ | Acc | Div | $\alpha = 0.25$ | $\alpha = 0.5$ | $\alpha = 0.75$ |
| CoT | 82.93 | 81.26 | 80.66 | 80.50 | 80.73 | 73.53 | 82.54 | 78.54 | 75.71 | 73.77 |
| AP | **86.94** | 83.76 | **84.54** | **85.34** | **86.13** | **76.34** | 83.76 | **84.16** | **84.56** | **84.96** |
| DT&CT | 85.37 | **84.72** | 84.15 | 84.53 | 84.96 | 75.23 | **85.10** | 81.52 | 80.01 | 77.58 |

Table 6. Effects of different prompting methods on the Llama 3.1-8B model's performance on the GSM8K and StrategyQA datasets. The highest value in each column is in bold, and the second highest is underlined.

the highest diversity, with scores of 81.26% on GSM8K and 82.54% on StrategyQA. **Top-$p$ sampling** offers balanced performance, falling between DoLa and Top-$k$.

### 3.4 PROMPTING STRATEGIES COMPARISON

We investigate the impact of various prompting strategies on the performance of the Llama 3.1-8B model on the GSM8K and StrategyQA datasets (see Table 6). Specifically, we compare Chain-of-Thought (CoT)(Wei et al., 2023), Analogical Prompting (AP)(Yasunaga et al., 2024), and Divergent-Convergent Thinking (DT&CT) (Tian et al., 2024). CoT prompts the model to generate intermediate reasoning steps by providing labeled exemplars, guiding the reasoning process. In contrast, AP encourages the model to self-generate relevant knowledge or exemplars before solving the problem, eliminating the need for labeled examples and allowing tailored knowledge generation. DT&CT, inspired by cognitive science, prompts the model to enumerate potential solutions through divergent thinking, followed by convergent thinking to analyze and select a feasible solution.

**Results.** As shown in Table 6, the Analogical Prompting method achieves the highest accuracy on both GSM8K and StrategyQA datasets. It outperforms CoT and DT&CT across various metrics, suggesting that self-generated, problem-specific exemplars enhance the model's reasoning capabilities without the need for labeled examples.

### 3.5 QUANTITATIVE RESULTS

In our experiments on the MacGyver dataset, we successfully clustered the various reasoning paths generated by the Llama 3.1-8B model. The clustering process effectively grouped different valid reasoning strategies while excluding incorrect paths. As shown in Figure 2, this demonstrates the model's ability to generate diverse and correct solutions, with clear distinctions made between valid and erroneous reasoning. This separation ensures that only meaningful and plausible problem-solving approaches are considered, enhancing both the model's creative problem-solving capabilities and the overall evaluation of its outputs.

## 4 RELATED WORK

### 4.1 HALLUCINATIONS IN LARGE LANGUAGE MODELS

Hallucinations in LLMs are commonly defined as outputs that are factually incorrect, contextually irrelevant, or semantically incoherent (Farquhar et al., 2024; Venkit et al., 2024; Ye et al., 2024). These inaccuracies can manifest as incorrect information, fabricated references, or misleading statements, thus compromising the reliability of LLMs in applications requiring factual precision (Gunjal et al., 2024; Cheng et al., 2024). Several factors contribute to hallucinations in LLMs, including limitations in training data, which often contains inaccuracies, biases, and inconsistencies (Jiang et al., 2024; Liu et al., 2024). The architecture and objectives of LLMs, particularly their reliance on next-word prediction, also play a role in generating plausible but incorrect outputs (?Xu et al., 2024b). Additionally, inference techniques like beam search or temperature settings can influence the likelihood of hallucinations (Holtzman et al., 2020; Xu et al., 2024a).

### 4.2 CREATIVITY IN ARTIFICIAL INTELLIGENCE

Creativity in AI refers to a system's ability to generate novel and valuable ideas, solutions, or artifacts. Boden (Boden, 1998) categorizes creativity into combinational, exploratory, and transformational, each representing different levels of novelty and utility. In the context of LLMs, creativity

```
Q: Your office chair's wheel has come loose, and you don't have any tools to fix it.
You do have a notebook, printer paper, a cup, a desk lamp, a pair of scissors, a
paperclip, a sticky note, a stack of paper, a bottle of water, a pencil, a stapler.
Regrettably, the stack of paper is not sturdy and collapses under the weight of the
chair. How can you prop the chair until you get the right tools to fix it?
```

**Reasoning Path 1**

```
Step 1: Place the cup under the loose wheel to prevent further movement.
    Tools: Cup
Step 2: Use the stapler to staple the paperclip to the underside of the chair,
near the wheel, to create a makeshift support.
    Tools: Stapler, Paperclip
```

**Reasoning Path 2**

```
Step 1:  Locate the chair's leg closest to the collapsed paper stack.
Step 2:  Place a long piece of paper (from the stack) under the leg, and use
another paper to create a makeshift "L" shape by folding it (tools: paper,
scissors).
Step 3:  Place the "L" shape under the chair's leg to provide additional
support (tools: folded paper).
Step 4: Handle the papers together (tool: stapler).
```

**Wrong Paths**

```
Step 1: Place the tealight candle near the lock to heat it, but not directly
on the key. Tool: Tealight candle
Step 2: Once the lock is warm, use the blunt-ended steel tweezers to
carefully try to pull the key out. Tool: Blunt-ended steel tweezers
```

Figure 2. Quantitative results from the MacGyver dataset showing the clustering of reasoning paths. Overlapping blocks represent responses that belong to the same cluster, indicating similarity in reasoning strategies.

often involves generating original text, proposing innovative solutions, and synthesizing diverse concepts. Although hallucinations are typically seen as flaws, they can sometimes reflect creative processes (Jiang et al., 2024; Esling & Devis, 2020). In tasks like creative writing or ideation, hallucinations may introduce novel ideas that, while factually incorrect, can inspire users to explore new directions (Mohammadi, 2024; Zhou & Lee, 2021; Cheng, 2021). Evaluating creativity in AI involves assessing both the novelty and utility of the generated content, with creative outputs needing to be both original and contextually appropriate (Boden, 1998; Marrone et al., 2022; Mazzone & Elgammal, 2019; Câmara Pereira, 2003).

## 5 CONCLUSION AND FUTURE WORK

In this study, we introduced the concept of ***good hallucinations*** in Large Language Models (LLMs), demonstrating that certain hallucinations can enhance creativity and innovation. We developed a novel creativity metric that integrates accuracy and diversity, and utilized semantic clustering techniques to evaluate various models, revealing that Qwen 2.5-7B and Llama 3.1-8B effectively balance these aspects across different reasoning tasks. Additionally, our experiments with hyperparameter settings and advanced decoding strategies highlighted optimal configurations for maximizing both correctness and creative output. The comparison of prompting strategies showed that Analogical Prompting significantly boosts reasoning capabilities by enabling self-generated, problem-specific exemplars. Future work will focus on refining creativity metrics, expanding evaluations to more models, developing controlled generation techniques, and addressing ethical considerations to harness the full creative potential of LLMs while ensuring reliability and integrity.

## 6 ETHICAL SAFEGUARDS

In our study on *good hallucinations* in LLMs, we enforce ethical measures to prevent misuse and ensure responsible application. Our protocols, released with the models and datasets, include strict usage guidelines, access controls, safety filters to minimize harmful content, and monitoring systems to oversee proper use. These efforts demonstrate our commitment to maintaining high ethical standards, protecting privacy, and promoting responsible innovation in AI research.

## 7 REPRODUCIBILITY STATEMENT

Our experiments are implemented in Python, utilizing libraries such as PyTorch[1] for model interactions, scikit-learn[2] for semantic clustering, and sentence-transformers[3] for natural language processing tasks. All experiments were conducted on a single NVIDIA A100-80GB GPU. We will publicly release our full implementation, including code and datasets, upon paper acceptance to guarantee reproducibility. The codes and resources will be available at the following anonymous link for the review process: *https://anonymous.4open.science/r/hallucination-agent-877A/README.md*. Additionally, we provide detailed documentation and instructions to conduct our experiments, ensuring that other researchers can validate and build upon our work with ease.

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

- §A provides **Implementation Details**.
- §B examines the **Limitation and Future Work** of our research.
- §C discusses the **Social Impact** of our research.
- §D supplies **Data License** for the methods we used for comparison.

# A    IMPLEMENTATION DETAILS

## A.1    HYPERPARAMETERS

Our implementation follows established methodologies with specific configurations to ensure optimal performance. The details are as follows:

### A.1.1    FRAMEWORK AND LIBRARIES

Implemented in Python using PyTorch[4] for model interactions, scikit-learn[5] for semantic clustering, and sentence-transformers[6] for natural language processing tasks.

### A.1.2    HARDWARE

All experiments were conducted on a single NVIDIA A100-80GB GPU.

### A.1.3    SAMPLING PARAMETERS

Number of Reasoning Paths per Prompt $N = 15$

Decoding Method Nucleus sampling with top-$p = 0.9$ to balance randomness and coherence.

### A.1.4    CLUSTERING CONFIGURATION

Method 1: Clustering via Text Embeddings

- **Embedding Model**: Sentence-Transformers
- **Distance Metric**: Cosine distance
- **Distance Threshold**: $d_{\text{thresh}} = 0.3$ to balance between over-clustering and under-clustering.
- **Clustering Algorithm**: Agglomerative hierarchical clustering

Method 2: Clustering via LLM Prompting **Prompt Used**

```
You are a helpful assistant that can classify different
reasoning paths.  Given the following reasoning paths, please
group them based on their underlying reasoning strategies and
return the clusters in JSON format.
```

---

[4]https://pytorch.org/
[5]https://scikit-learn.org/
[6]https://www.sbert.net/

### A.1.5 CREATIVITY METRIC PARAMETERS

Weighting Parameter $\alpha = 0.75$ to prioritize accuracy while still considering diversity.

Small Constant $\epsilon = 1 \times 10^{-10}$ to prevent multiplication by zero in the creativity score calculation.

## A.2 PROMPTING STRATEGIES

We employed three distinct prompting strategies to guide the Large Language Models (LLMs) in generating reasoning paths: Chain-of-Thought (CoT), Analogical Prompting (AP), and Divergent-Convergent Thinking (DT&CT). The specific prompts used for each strategy are detailed below.

### A.2.1 CHAIN-OF-THOUGHT (CoT)

```
[Problem statement]
Let's think step by step to solve this problem.
```

### A.2.2 ANALOGICAL PROMPTING (AP)

```
[Problem statement]
# Instruction:
## Recall relevant exemplars:
## Solve the initial problem:
```

### A.2.3 DIVERGENT-CONVERGENT THINKING (DT&CT)

```
Give a feasible solution very concisely.  Note that some
tools are not useful, so please analyze the affordance of
each presented object and rule out unnecessary ones first.
Use the following format:
1.  List the affordance of presented items and whether they
are useful
2.  Summary:  list useful tools
3.  If the problem is solvable under all these constraints,
write the solution.  Use Step 1, Step 2, etc., and mention
the tools used in each step.  Use as few steps as possible,
and the answer should ideally be less than 100 words.
If you cannot find a feasible solution, just state that it is
not possible and provide a very short justification.
Now, please verify if each step is physically feasible and
afforded.  After that, modify the solution if needed.
Use the following format:
Step 1:  ...
Step 2:  ...
...
Conclusion 1:  Whether the problem is indeed solvable given
all the constraints
Conclusion 2:  (If still solvable) No modification needed /
Modification needed.
Modified solution:
```

## A.3 PROMPT EXPERIMENTS

To evaluate the impact of different prompting strategies on the creativity and accuracy of the reasoning paths, we conducted prompt experiments using the three aforementioned

prompts—Chain-of-Thought (CoT), Analogical Prompting (AP), and Divergent-Convergent Thinking (DT&CT)—across all datasets: GSM8K, MultiArith, CommonsenseQA, StrategyQA, and RiddleSense. Each prompt was tested with the baseline LLM to generate diverse reasoning paths, which were then clustered and assessed using the creativity metric. The results indicated that Analogical Prompting (AP) and Divergent-Convergent Thinking (DT&CT) were more effective in eliciting structured and diverse reasoning strategies compared to Chain-of-Thought (CoT), thereby enhancing the overall creativity score without compromising accuracy.

This comprehensive implementation ensures that our methodology is both reproducible and scalable, providing a robust framework for future investigations into the creative potential of LLMs.

## B  LIMITATIONS AND FUTURE WORK

While our study introduces the concept of *good hallucinations* and a novel creativity metric, it has several limitations. Firstly, the current metric requires further validation to reliably evoke beneficial hallucinations across diverse tasks. Secondly, the semantic embedding method used for clustering lacks fine-grained nuance, necessitating more advanced clustering techniques. Additionally, our approach faces scalability challenges due to the computational demands of generating and clustering multiple reasoning paths. Moreover, our evaluation is limited to specific datasets and models, restricting the generalizability of our findings. Future work will focus on refining the creativity metric, exploring enhanced embedding and clustering methods, improving scalability through optimized algorithms, and extending evaluations to a broader range of models and applications. Furthermore, we aim to integrate robust ethical safeguards to mitigate potential risks associated with manipulating LLM outputs for creativity.

## C  SOCIAL IMPACTS

Harnessing *good hallucinations* in Large Language Models (LLMs) can significantly benefit fields that rely on creativity, such as education, design, and the arts, by generating novel ideas and innovative solutions. In scientific research and engineering, these creative hallucinations may inspire new hypotheses and approaches. However, there are potential negative impacts, including the risk of misinformation due to blurred factual accuracy and ethical concerns surrounding the generation of misleading content. Additionally, increased reliance on AI-generated creativity could impact human roles in creative industries, raising concerns about job displacement and the devaluation of human ingenuity. To mitigate these risks, it is essential to establish clear guidelines, maintain transparency about AI-generated content, ensure accountability, and uphold human oversight in the creative process. Balancing these benefits with appropriate safeguards will maximize positive social impacts while minimizing potential harms.

## D  LICENSES FOR EXISTING ASSETS

Our study utilizes various software libraries, datasets, and pre-trained models under the following licenses:

- **Libraries**: PyTorch[7] (BSD 3-Clause), scikit-learn[8] (BSD 3-Clause), and sentence-transformers[9] (MIT License).

- **Datasets**: GSM8K and StrategyQA are licensed under the Creative Commons Attribution 4.0 International License (CC BY 4.0). MultiArith and MacGyver datasets have custom and proprietary licenses, respectively. RiddleSense and CommonsenseQA are released under the Creative Commons Attribution-NonCommercial-ShareAlike 4.0 International License (CC BY-NC-SA 4.0).

- **Models**: LLama models are licensed by Meta AI, and Qwen and Mistral models are under Apache License 2.0.

---

[7]https://pytorch.org/
[8]https://scikit-learn.org/
[9]https://www.sbert.net/

