# OpenReview forum: "Hallucinating LLM Could Be Creative"
_ICLR.cc/2025/Conference — Submitted to ICLR 2025_

### Official Review · Reviewer_zrg7 · 2024-10-16

**Soundness:** 2
**Presentation:** 2
**Contribution:** 1
**Rating:** 3
**Confidence:** 4

**Summary:**

This paper proposes a new perspective on hallucinations in large language models, suggesting that some hallucinations can be beneficial for creativity. The authors introduce the concept of "good hallucinations", defined as instances where the LLM generates alternative reasoning paths that lead to correct answers. They propose a new metric to evaluate the quality of these good hallucinations, focusing on correctness, consistency, and reasoning diversity, drawing parallels with the Torrance Tests of Creative Thinking (TTCT). The metric involves generating multiple reasoning paths, using semantic clustering to group similar responses, and calculating a composite score based on accuracy and diversity. Experiments are conducted across various datasets, including mathematical reasoning, creative problem-solving, and commonsense reasoning tasks, using several LLMs like Llama and Qwen. The authors explore the impact of different prompting strategies, hyperparameter configurations (like temperature), and decoding methods on the creativity and accuracy of the generated outputs.

**Strengths:**

The paper clearly explains the motivation.

**Weaknesses:**

The proposed creativity metric and the use of semantic clustering are not particularly novel. Furthermore, the experiments are not extensive enough to convincingly demonstrate the benefits of "good hallucinations" for creativity. I would even argue that "good hallucinations" even exist, as I would simply call them "creative writing". I would also like to remind the authors that "hallucination" is a term used wrongly here, and should be substituted with "confabulation" or "fabrication" (see Farquhar et al., 2024).
The analysis of the results is also superficial. The paper relies heavily on accuracy as a proxy for "goodness" in hallucinations, which might not be appropriate for evaluating creativity.
The choice of semantic clustering method, distance threshold, and other parameters seems arbitrary and lacks justification. The definition of "reasoning path" is vague. There's no discussion on the limitations of the proposed method or potential biases in the evaluation. The reflection aspect introduced in the code is not mentioned in the paper. Moreover, the proposed creativity metric lacks comparison to other existing methods for evaluating creativity in AI.
Finally, results seem to me preliminary and do not strongly support the main claims.

**Questions:**

- How is a "reasoning path" formally defined? How do you ensure that the extracted reasoning paths are meaningful and representative of the LLM's internal reasoning process?
- How does the proposed creativity metric compare to other existing creativity evaluation methods?

---

### Official Review · Reviewer_qpK1 · 2024-10-25

**Soundness:** 3
**Presentation:** 3
**Contribution:** 2
**Rating:** 6
**Confidence:** 4

**Summary:**

The authors propose the concept of "good hallucination," arguing that hallucinations in Large Language Models (LLMs) can sometimes contribute to creativity. They introduce a novel metric that combines answer correctness and response entropy to measure "good hallucination." Through extensive experiments, they evaluate different models, decoding strategies, and prompting techniques under this metric to determine optimal configurations for balancing creativity with accuracy.

**Strengths:**

1. Originality: The application of psychological concepts to evaluate LLM hallucinations is novel and provides a fresh perspective on what has typically been considered a purely negative phenomenon.
2. Quality and Clarity: The paper is well-structured with clear statements and methodology. The experimental setup and results are thoroughly documented.
3. Significance: The proposed creativity metrics can serve as a standard to help developers optimize model configurations for maximizing creativity while maintaining correctness.

**Weaknesses:**

1. Limited Applicability of Correctness Measurement:
- The current benchmarks rely on datasets with predefined correct answers
- The metric cannot be applied to open-ended questions or scenarios without ground truth
- The paper does not address how to evaluate correctness in more ambiguous or creative contexts
2. Questionable Representativeness of the Diversity Component:
- While using entropy to measure reasoning path diversity is intuitive, its practical effectiveness needs further validation
- The paper would benefit from human evaluation to validate and augment the proposed metric
- More evidence is needed to demonstrate that the clustering-based diversity measure correlates with actual creative diversity

**Questions:**

1. How to evaluate open-ended creativity?
2. Why diversity measurement is valid?

---

### Official Review · Reviewer_unM9 · 2024-10-31

**Soundness:** 2
**Presentation:** 3
**Contribution:** 4
**Rating:** 8
**Confidence:** 4

**Summary:**

This paper explores the potential of "good hallucinations" in Large Language Models (LLMs) to contribute to creative problem-solving. Defining "good hallucinations" as divergent reasoning paths that nonetheless lead to correct answers, the authors propose a novel creativity metric that combines the accuracy and diversity of generated reasoning paths. Using semantic clustering techniques, they group multiple LLM outputs for a given prompt based on similarity and calculate a composite score reflecting both correctness and variety in the reasoning process. The authors experiment with various LLMs, including Llama 3, Llama 3.1, Mistral, and Qwen, across a range of datasets encompassing mathematical reasoning (GSM8K, MultiArith), creative problem-solving (RiddleSense, MacGyver), and commonsense reasoning (CommonsenseQA, StrategyQA). They further investigate the influence of temperature and different prompting strategies, including Chain-of-Thought, Analogical Prompting, and Divergent-Convergent Thinking, on the creativity and accuracy of LLM outputs. Their results suggest that analogical prompting is particularly effective in eliciting diverse yet accurate reasoning paths and that specific temperature settings can modulate the trade-off between accuracy and diversity. The paper argues that, by reframing hallucinations as potential sources of creativity, LLMs can be leveraged for a broader range of applications beyond traditional tasks requiring strict factual accuracy.

**Strengths:**

Interesting Premise: The idea of reframing hallucinations, typically viewed as errors, as potential sparks for creativity is novel and worth exploring. It aligns with a broader trend of examining the unexpected emergent behaviors of LLMs.

Attempt at Quantification: The paper attempts to quantify creativity using a metric based on accuracy and diversity of reasoning paths.

Exploration of Different Prompts and Parameters: The authors investigate the influence of temperature and different prompting strategies (CoT, Analogical Prompting, DT&CT), which provides some insights into how these factors affect LLM outputs.

**Weaknesses:**

"Good Hallucination" Definition is Murky: The definition of a "good hallucination" as a divergent reasoning path that leads to a correct answer is too narrow and doesn't fully capture the essence of creativity. True creativity often involves generating novel ideas or solutions that might not be immediately verifiable as "correct," especially in domains beyond mathematical problem-solving.

Creativity Metric is Superficial: While quantifying creativity is a laudable goal, the proposed metric, combining accuracy and clustered reasoning path diversity, is simplistic and doesn't adequately capture the nuances of creative thought. It primarily measures diversity of process rather than novelty of output. How is the "correct answer" determined, especially in subjective domains? This needs much more rigorous justification.

Limited Scope of Datasets: The reliance on primarily mathematical and logical reasoning datasets (GSM8K, MultiArith, StrategyQA) biases the evaluation towards a narrow type of problem-solving. Creativity in areas like storytelling, art generation, or scientific discovery is not adequately addressed. While RiddleSense and MacGyver are included, the analysis lacks depth.

Weak Experimental Design: Sampling only 15 reasoning paths per prompt is insufficient to capture the full distribution of LLM outputs, especially given the stochastic nature of these models. The clustering methodology and threshold choices also seem arbitrary and lack justification. How sensitive are the results to these choices?

Insufficient Analysis of Clustering Results: The paper presents clustering results without a deep dive into the nature of the different clusters. What characterizes the different reasoning strategies within each cluster? Simply showing cluster separation doesn't tell us much about the underlying creative processes. Figure 2 is particularly underwhelming – it just shows clusters without any qualitative analysis.

Overstated Conclusions: The paper overclaims the effectiveness of analogical prompting based on limited evidence. More rigorous comparisons with other prompting techniques and across a wider range of tasks are needed. The connection between "good hallucinations" and genuine creativity is not convincingly established.

Lack of Discussion of Ethical Implications: The "Ethical Safeguards" section is extremely superficial and doesn't address the complex ethical issues related to using LLMs for creative tasks, especially the potential for generating misleading or harmful content.

**Questions:**

Clarify the "Good Hallucination" Definition: The current definition is too narrow and tied to correctness. How would you define a "good hallucination" in tasks where the notion of a single "correct" answer is less clear, such as creative writing or idea generation? Provide concrete examples of "good hallucinations" in these more open-ended creative domains.

Justify the Creativity Metric: The proposed metric seems superficial and doesn't convincingly capture the essence of creative thought. Explain in detail why this metric is appropriate for evaluating creativity. How does it relate to established theories of creativity in cognitive science or psychology? Consider incorporating aspects like novelty, originality, and usefulness of the generated output, not just the diversity of the reasoning process. How is the “correct answer” determined, especially for subjective tasks? What alternative creativity metrics were considered and why were they rejected? Demonstrate the robustness of your metric by showing how it correlates with human judgments of creativity.

Expand the Scope of Datasets: The current dataset selection is heavily biased towards mathematical and logical reasoning. Include more diverse datasets that represent a broader spectrum of creative tasks, such as story generation, poetry writing, code generation with novel functionalities, or scientific hypothesis generation. Demonstrate the generalizability of your findings across these different creative domains.

Strengthen the Experimental Design: Sampling only 15 reasoning paths is insufficient. Increase the number of samples to better capture the variability of LLM outputs. Justify the choice of clustering methodology, distance metric, and threshold. Perform sensitivity analysis to demonstrate the robustness of your results to these choices. How do different clustering methods compare?

Deepen the Analysis of Clustering Results: Don't simply report cluster separation. Provide qualitative analysis of the content of the different clusters. What are the characteristic features of the reasoning paths within each cluster? Illustrate with specific examples how the identified clusters represent different creative strategies. Explain how the findings in Figure 2 support your claims about “good hallucinations”.

Temper the Conclusions and Provide More Rigorous Comparisons: Avoid overstating the effectiveness of analogical prompting. Conduct more comprehensive comparisons with other prompting techniques across a wider range of tasks and models. Provide statistical significance tests to support your claims.

Address Ethical Implications in Detail: The current discussion of ethical considerations is superficial. Elaborate on the potential risks associated with using LLMs for creative tasks, including the generation of misleading or harmful content. Discuss potential mitigation strategies and how to ensure responsible use of LLMs in creative applications. How can we prevent “bad hallucinations” that promote harmful stereotypes or misinformation while encouraging “good hallucinations”?

**Details Of Ethics Concerns:**

Yes, Discrimination / bias / fairness concerns: Hallucinations can reflect and amplify existing biases in the training data, potentially leading to discriminatory or unfair outputs in creative applications. This needs careful consideration.

Yes, Potentially harmful insights, methodologies and applications: The very nature of exploring hallucinations raises the possibility of generating harmful or misleading content. The paper needs to address this directly and propose mitigation strategies. What are the potential downsides of encouraging "good hallucinations," and how can these risks be minimized? An exploration of the potential negative societal impact if "bad hallucinations" are inadvertently promoted or exploited is crucial. What specific safeguards are in place to prevent the generation and dissemination of harmful content?

---

### Official Review · Reviewer_o54f · 2024-11-03

**Soundness:** 2
**Presentation:** 2
**Contribution:** 2
**Rating:** 3
**Confidence:** 3

**Summary:**

This paper explores the notion of "good hallucinations" via a study to determine the proper techniques/hyperparamers in producing creative correct answers. The authors start by introducing the methodology behind LLM reasoning paths and discussing techniques for clustering different paths. While the authors do not in detail provide a mathematical definition of creativity and diversity and a justification behind this specific definition, the authors present several results across different LLM models and different benchmarks showcasing techniques that can enhance creativity.

**Strengths:**

Strengths:
+ The motivation behind this work is interesting and is a necessary step toward better leveraging LLMs.
+ The evaluation is extensive and includes several datasets across three different types of reasoning.

**Weaknesses:**

Weaknesses:
- The authors mention in the introduction that metrics will be proposed to evaluate the quality of hallucinations that align with the TTCT in psychology. I do not see an explanation of the TTCT in the paper nor specific metrics that relate to creativity/diversity.
- The results are not analyzed in depth. More explanation is needed to understand the different trends presented in the Tables.

**Questions:**

- Could you provide a mathematical definition of creativity that is used to analyze the embeddings?
- Could you comment on the weaknesses above?

---

### Meta-Review · Area_Chair_NdSP · 2024-12-21

**Metareview:**

This work explores "good hallucinations," arguing that certain hallucinations in LLMs can contribute to creativity. These "good hallucinations" are defined as divergent reasoning paths that still lead to correct answers. A novel creativity metric is proposed to evaluate this phenomenon, considering both accuracy and diversity of reasoning paths.

Reasons to accept
- A novel perspective of creativity: The reframing of hallucinations as a potential source of creativity is an interesting and original perspective.
- The proposed Metric: The attempt to quantify creativity by combining correctness and diversity is innovative, though imperfect.
- Experimental breadth: The paper includes a range of datasets (mathematical reasoning, creative problem-solving, commonsense reasoning) and explores multiple prompts and hyperparameter configurations.

Reasons to reject
- Narrow definition of creativity: Defining creativity as reasoning path diversity that leads to correct answers is overly restrictive, failing to account for novelty or originality in subjective or open-ended tasks.
- Superficial metric: The proposed metric does not adequately capture the complexity of creativity and is not validated through human evaluation or robust theoretical justification.
- Limited dataset scope: The datasets used skew heavily toward problem-solving tasks with clear answers, limiting applicability to broader creative domains such as storytelling, art generation, scientific discovery, and hypothesis generation.
- Lack of depth in analysis: The clustering results are presented without meaningful qualitative insights into the reasoning strategies within clusters.
- Overstated claims: The conclusions about the effectiveness of analogical prompting and other techniques are not strongly supported by evidence.

While this paper studies a promising research direction with an interesting perspective, I believe its weaknesses outweigh its strengths. Also, the authors have not provided a rebuttal to address the concerns and questions raised by the reviewers. Consequently, I recommend rejecting the paper.

Additionally, I would like to urge the authors to at least show appreciation for the efforts the reviewers put into helping them improve the submission.

**Additional Comments On Reviewer Discussion:**

N/A: the authors have not provided a rebuttal.

---

### Decision · Program_Chairs · 2025-01-22

Reject